# Ferroptosis and Apoptosis Are Involved in the Formation of L-Selenomethionine-Induced Ocular Defects in Zebrafish Embryos

**DOI:** 10.3390/ijms23094783

**Published:** 2022-04-26

**Authors:** Meng Gao, Jun Hu, Yuejie Zhu, Xianqing Wang, Shumin Zeng, Yijiang Hong, Guang Zhao

**Affiliations:** School of Life Science, Nanchang University, Nanchang 330031, China; gaomeng@webmail.ncu.edu.cn (M.G.); hu13404032256@163.com (J.H.); zhuyjjane@163.com (Y.Z.); wangxianqing2021@163.com (X.W.); shuminzeng@email.ncu.edu.cn (S.Z.)

**Keywords:** L-selenomethionine, ROS, apoptosis, ferroptosis, microphthalmia

## Abstract

Selenium is an essential trace element for humans and other vertebrates, playing an important role in antioxidant defense, neurobiology and reproduction. However, the toxicity of excessive selenium has not been thoroughly evaluated, especially for the visual system of vertebrates. In this study, fertilized zebrafish embryos were treated with 0.5 µM L-selenomethionine to investigate how excessive selenium alters zebrafish eye development. Selenium-stressed zebrafish embryos showed microphthalmia and altered expression of genes required for retinal neurogenesis. Moreover, ectopic proliferation, disrupted mitochondrial morphology, elevated ROS-induced oxidative stress, apoptosis and ferroptosis were observed in selenium-stressed embryos. Two antioxidants—reduced glutathione (GSH) and N-acetylcysteine (NAC)—and the ferroptosis inhibitor ferrostatin (Fer-1) were unable to rescue selenium-induced eye defects, but the ferroptosis and apoptosis activator cisplatin (CDDP) was able to improve microphthalmia and the expression of retina-specific genes in selenium-stressed embryos. In summary, our results reveal that ferroptosis and apoptosis might play a key role in selenium-induced defects of embryonic eye development. The findings not only provide new insights into selenium-induced cellular damage and death, but also important implications for studying the association between excessive selenium and ocular diseases in the future.

## 1. Introduction

Selenium is an essential microelement required by humans and other vertebrates for several important biological functions [1]. The cellular effects of selenium are highly concentration-dependent, as either too low or too high selenium concentration can cause cell death [2]. At moderate concentrations, selenium has antioxidant activity with certain protective effects [3,4], while at high levels, it displays prooxidant activity with some toxic effects. Excessive selenium is associated with many human diseases, such as type 2 diabetes, high-grade prostate cancer, amyotrophic lateral sclerosis and Parkinson’s disease [5,6,7,8]. In the aquatic environment, selenium mainly exists in inorganic forms such as selenate and selenite [9], and can be readily transformed into an organic selenium form (selenomethionine) by primary producers and certain microorganisms, which may lead to bioaccumulation and biomagnification through incorporation into aquatic food webs [10]. Thus, selenomethionine is the predominant source of available selenium for fish, particularly in selenium-contaminated aquatic environments with selenium concentrations ranging from 0.13 µM to 0.86 µM [11,12]. Selenomethionine is also the main existing form of selenium in animal and human tissues, where it can be incorporated nonspecifically with the amino acid methionine in body proteins [13]. Since inorganic and organic species have distinct biological properties, different forms of selenium may have different levels of toxicity.

The visual system is an increasingly recognized sensitive target of aquatic pollutants. Various environmental pollutants of distinct physicochemical properties can accumulate in the eye and impair the eye development and function in teleost fishes [14]. Excessive selenium may increase the risk of cataracts, glaucoma, ocular hypertension and damage to the conjunctiva and cornea [15]. Many environmental pollutants have been reported to induce the production of reactive oxygen species (ROS) in vertebrates [16]. Excessive selenium has also been reported to induce massive production of ROS and cause pronounced oxidative stress [17]. High intracellular levels of ROS can induce pathology by damaging lipids, proteins and DNA, alter signaling pathways and influence genetic and epigenetic changes in cells [18,19,20]. In order to maintain redox homeostasis, cells in the body will regulate a series of related genes to remove ROS or repair ROS-induced damage to proteins, lipids and nucleic acids under oxidative stress [21,22]. The selenium form and/or cell line can determine the activation of different intracellular signaling pathways that lead to cell death. For example, sodium selenite induces necroptosis-like cell death, whereas selenodiglutathione leads to apoptosis-like cell death in HeLa cells [2,17]. Treatment of human glioma cell lines with sodium selenite can induce apoptosis [23]. Excessive sodium selenite also causes cytotoxicity and apoptosis-mediated cell death in the PLHC-1 fish cell line by damaging the DNA and mitochondrial membrane potential [24]. Many eye diseases, such as diabetic retinopathy, glaucoma, keratopathies, cataract formation and age-related macular degeneration (AMD), are associated with oxidative stress arising from the ubiquitous production of ROS [25]. ROS play important roles in regulating numerous physiological activities and cellular metabolism in normal cells, such as vascular reactivity and neuron function. However, excessive ROS have been associated with vascular endothelial dysfunction, neuron degeneration and inflammation in the retina [26,27,28]. ROS-induced apoptosis in human lens epithelial cells is related to autosomal dominant congenital cataract (ADCC), a common hereditary disease [29]. Oxidative stress has been found to cause ferroptosis in retinal pigment epithelial cells [30]. However, the toxic effect of excessive selenomethionine on eyes remains unknown. Therefore, we hypothesize that excessive selenomethionine can cause ocular defects by inducing oxidative stress through ROS accumulation.

Zebrafish eye development is highly similar to that of humans and other vertebrates, being of high molecular complexity and stringent spatiotemporal regulation including the surface ectoderm, neuroectoderm and mesenchyme [31,32]. Therefore, in order to reveal the potential mechanism for selenomethionine to induce eye deformity, zebrafish embryos were treated with selenomethionine (0.5 µM) in this study as previously described [33]. The eye radius of zebrafish embryos was measured and recorded first. Then, the histological changes in the eyes of selenium-stressed embryos were examined by hematoxylin and eosin (H&E) staining and transmission electron microscopy (TEM), followed by the determination of the expression of genes involved in eye development. Then, the intracellular ROS, redox enzymes and expression of oxidative stress marker genes were measured. In addition, the modes of cell death in selenium-stressed embryonic eyes were identified by qPCR, TUNEL assay and FerroOrange staining. Finally, two antioxidants—reduced glutathione (GSH) and N-acetylcysteine (NAC)—along with a ferroptosis inhibitor, ferrostatin-1 (Fer-1), and a ferroptosis and apoptosis activator, cisplatin (CDDP), were tested for their effects to alleviate selenium-induced ocular defects.

## 2. Materials and Methods

### 2.1. Fish, Reagents and Antibodies

The AB wild-type (WT) adult zebrafish were maintained in a circulation filtration system (28 ± 0.5 °C, pH 7.0–8.0, conductivity of 500–800 µS/cm) with a light:dark ratio of 14:10 h and fed with hatched fairy shrimp three times per day. Embryos were obtained by natural spawning and cultured at 28.5 °C in an incubator. The embryonic and larval ages were expressed as hours post-fertilization (hpf) or days post-fertilization (dpf). The reagents and antibodies used in the study were as follows: L-selenomethionine (Sigma-Aldrich, St. Louis, MO, USA), TUNEL Bright-Red Apoptosis Detection Kit (Vazyme, Nanjing, China), DAPI, FerroOrange (Sigma-Aldrich, St. Louis, MO, USA), NAC, Reduced GSH, Fer-1 and CDDP (MCE, Monmouth Junction, NJ, USA). Antibodies used were phospho-H3 (PH-3) (Cell Signaling Technology, Danvers, MA, USA), Caspase3 and FITC Goat Anti-Rabbit IgG (ABclonal, Wuhan, China).

### 2.2. Chemical Treatment and Phenotype Observation

According to our previous study, selenomethionine at the concentration of 0.5 µM was used for all selenium stress experiments [33]. First, 39.5 mg selenomethionine was dissolved in 1 L distilled water to obtain a stock solution with 0.5 mM concentration, and then the stock solution was diluted to the desired concentration of 0.5 μM in embryo medium. Zebrafish embryos (before 2 hpf) were distributed into a 60 mm diameter Petri dish containing 60 embryos and 10 mL selenomethionine solution with 0.5 μM concentration, and then put in a 28 ± 0.5 °C incubator for selenium exposure lasting until 96 hpf. During the exposure, the dead embryos were discarded every day, and the solution was not replaced. The embryos of different timepoints (24, 48, 72 and 96 hpf) were collected according to our experimental requirements. Embryos from the same batch were used for treatment groups and control groups in each experiment. Each experiment was replicated three times. The zebrafish embryos were anesthetized and photographed at 24, 48, 72 and 96 hpf to observe the eye developmental defects induced by selenium. The eye radius (µm) of about 30 zebrafish embryos from both the control and treatment groups was measured and recorded using a microscope (SMZ1270, Nikon, Tokyo, Japan) at 48, 72 and 96 hpf. NAC (200 µM), GSH (100 µM), Fer-1 (120 nM) and CDDP (10 µM) were added after selenium treatment.

### 2.3. RNA-seq

Embryos in the 0.5 µM selenomethionine-exposed group and the control group at 96 hpf were collected and used for RNA extraction and RNA-Seq analysis. RNA-Seq of different samples (Both selenium and control groups have three replicates) was performed in Novogene Co. Libraries were constructed and sequenced using an Illumina HiSeq^TM^. Normalized read counts were used for fold change analysis, and fold changes (FC) were determined for each gene through dividing the number of tags in the normalized selenomethionine treatment libraries by that in the normalized control libraries by DEGSeq. Genes that were significantly altered as a result of the selenomethionine treatment (adjusted *p* < 0.05) were defined as differentially expressed genes (DEGs) and were used for further analysis.

### 2.4. Quantitative PCR

Extraction of total RNA was performed by using Trizol reagent (TAKARA, Kusatsu, Japan) following the manufacturer’s protocol. RNA was quantified by using a Jenway™ Genova Nano Micro-volume Spectrophotometer (Bibby Scientific Ltd., Stone, Staffs, UK). RNA integrity was evaluated using agarose gel electrophoresis stained with GelRed (Appendix A). RNA was reverse transcribed to cDNA with an PrimeScript™ RT reagent Kit (TAKARA) according to the manufacturer’s protocol. The real-time PCR with 20 ng of cDNA/reaction in a final volume of 20 µL was performed in a CFX96^TM^ Real-Time System (Bio-Rad, Hercules, CA, USA) using TB Green™ Premix Ex Taq™ II (TAKARA). β-actin was used for the relative quantification of within-reaction expression. The Ct values were obtained with the CFX Manager software (Bio-Rad, Hercules, CA, USA) and the data were then employed to analyze the gene expression levels. Three technical replicates were used for each sample, and the average of three biological replicates was calculated.

### 2.5. Whole-Mount in Situ Hybridization (WISH)

Whole-mount in situ hybridization (WISH) was performed as described previously [34]. The primers used for probe synthesis were as follows:

*opn1sw1* (F: TTTGGATGGAGCAGATAC, R: TGAGAAGAAAGCAGGGAT),

*rx1* (F: GTGCAGGTTTGGTTCCAG, R: CTCCAGAGGGTATTTGTCG),

*cryaa* (F: CACCCTTGGTTCAGACGC, R: AGGAAGAGCCCGAGTTGC),

*pde6a* (F: GACAAGGATGTGGCTGAG, R: AGGGTAGCGTCTTTATGG),

*cox4i2* (F: AGGCAGATATGTCTCGACCAATG, R: TGTTCCAGGGCCCTTTCTCC),

*nrf2* (F: TCTGCCCGTAGACGACTTT, R: ACTCCAATCCCACGATGTT),

*otpa* (F: GCGTCCAGGTTTGGTTTC, R: GGGAAGGTGGGTTGGTAGA).

### 2.6. Intracellular ROS and Redox Enzymes Detection

Intracellular ROS levels in embryonic eyes were measured using a DCFH-DA (2′, 7′-Dichlorodihydrofluorescein-diacetate) Reactive Oxygen Species Assay Kit (Beyotime, Shanghai, China) according to the manufacturer’s instructions. The embryos were photographed under a microscope (SMZ1270, Nikon, Japan). The activity or content of malondialdehyde (MDA, A003-2), glutathione reductase (GR, A062-1-1) and GSH (A006-2-1) was detected with Nanjing Jiancheng Bioengineering Institute’s commercial kits (Nanjing Jiancheng Bioengineering Institute, Nanjing, China) following the manufacturer’s protocol.

### 2.7. Transmission Electron Microscopy (TEM)

The zebrafish embryos, including the control and 0.5 µM selenomethionine-stressed embryos, were collected at 96 hpf, cleaned with dH_2_O twice and prefixed in 4% glutaraldehyde solution (pH 7.4) at 4 °C for more than 2 h. Then, the embryos were fixed with 1% osmium tetroxide at room temperature (20 °C) for 2 h, followed by ethanol gradient dehydration. Next, the embryos were embedded in Quetol 812 (Nisshin EM Co., Ltd., Tokyo, Japan) for sectioning. After double staining with uranium acetate and lead citrate, a transmission electron microscope (Hitachi H-7650 TEM, Tokyo, Japan) was used to observe and photograph the embryos. The structure of organelle mitochondria in eye cells of both the control and treatment groups was tested.

### 2.8. Cryosection

Zebrafish embryos were collected at 48, 72 and 96 hpf and fixed with 4% paraformaldehyde (PFA) overnight at 4 °C. Before cryosectioning, embryos were dehydrated with 30% sugar solution for at least 4 h at 4 °C. Next, the embryos were embedded in Tissue-Tek^®^ O.C.T. compound (Sakura Finetek, Torrance, CA, USA) for cryosectioning at 8 μm in thickness with frozen microtomy (Thermo Scientific, Waltham, MA, USA). Then, the samples were attached to the adhesion microscope slides (Citotest, Haimen, China). After drying at 37 °C for 30 min, the sections were stored and later used for hematoxylin and eosin (H&E) staining, FerroOrange staining, immunofluorescence and TUNEL assay.

### 2.9. Immunofluorescence and Apoptosis Assays

Immunofluorescence was performed with frozen sections. After two washes with PDT for 10 min per time, the sections were sealed with sealing fluid (PBS added with 2% BSA) at room temperature for 1 h. Then, the primary antibody against PH-3 and Caspase3 at 1/200 dilution was added, followed by incubation overnight at 4 °C. Next, after washing three times with PDT for 10 min per time, the sections were incubated with the secondary antibody conjugated with Alexa Fluor 488 at a 1:1000 dilution at 37 °C for 1 h. Afterwards, 4′, 6′-diamidino-2-phenylindole (DAPI) was used to label the nuclei. The sections were washed three times with PBS for 15 min per time. A TUNEL Detection Kit (Vazyme, Nanjing, China) was used to detect the apoptosis following the protocol. The images of immunofluorescence and apoptosis were captured using a confocal microscope (Nikon, Japan).

### 2.10. FerroOrange Staining

FerroOrange (Sigma-Aldrich, St. Louis, MO, USA) was used to determine the Fe^2+^ level in embryonic eyes of both the control and treatment groups. The cryosections were gently rinsed for twice with PDT for 5 min to remove extracellular Fe^2+^. Afterwards, 1 mM stock solution of FerroOrange was diluted in PBS to prepare a staining solution with a final concentration of 1 µM. The staining solution was added to cover the samples. Then, the frozen sections were put into a wet box and incubated at 37 °C for 30 min. DAPI was used to label the nuclei for 5 min. After the staining, the stained cells were rinsed twice with PBS for 5 min per time and observed with a confocal microscope (Nikon, Japan).

### 2.11. Statistical Analysis

The sample size used for different experiments in each group was larger than 10 embryos (*n* > 10), and 3 biological replicates were performed for each test. GraphPad Prism version 9.0 was used to visualize the statistical data. The figure of eye radius was drawn by SuperPlots in GraphPad Prism according to the previous study [35]. The signal area of WISH, intensity of FerroOrange and expression of Caspase3 were calculated by Image J version 1.53k. The statistical significance of eye radius, MDA and GSH content, GR activity, signal area of WISH, intensity of FerroOrange and qPCR results among multiple groups were analyzed by one-way analysis of variance (ANOVA) and post hoc Tukey’s test on GraphPad Prism version 9.0. The statistical significance of other data compared between two groups were analyzed by Student’s unpaired 2-tailed *t*-test on GraphPad Prism version 9.0. All data were expressed as mean ± SEM, and statistically significant differences among groups were indicated by *p* < 0.01 (**) and *p* < 0.05 (*).

## 3. Results

### 3.1. Excessive Selenium-Induced Ocular Defects in Zebrafish Embryos

Morphological and histological observations were conducted to examine the developmental abnormality induced by excessive selenium in zebrafish embryos at 24, 48, 72 and 96 hpf. As shown in Appendix A, selenium-stressed embryos had eye hypoplasia, yolk sac edema, short body length and pericardial edema, and some embryos showed developmental delay. As for the eye radius, selenium-stressed embryos exhibited a reduction in eye radius at 48, 72 and 96 hpf (Figure 1A). In addition, H&E staining analysis showed obvious ocular damage in embryos at 96 hpf (Figure 1B). RNA-seq was used to determine the gene expression profiles in selenium-stressed embryos. All the DEGs are illustrated in Appendix A. Gene ontology (GO) and Kyoto Encyclopedia of Genes and Genomes (KEGG) pathway analyses were performed to understand the interactions of the DEGs. All these analysis results of GO and KEGG are illustrated in Appendix A. The GO enrichment analysis revealed that the DEGs were enriched in eye development, retinol metabolism, retinal ganglion cell axon guidance, visual perception, sensory perception of light stimulus and response to light stimulus (Appendix A). KEGG pathway enrichment analysis indicated that the DEGs were enriched in retinol metabolism and phototransduction (Appendix A). All the expressions of eye-related genes enriched by GO and KEGG are shown in heatmaps (Appendix A).

To identify the toxic effect of excessive selenium on zebrafish embryonic eyes, qPCR and WISH were performed to test the expression of the genes related to eye development in both the control and treated embryos at 24 hpf and 96 hpf. The results revealed that among the tested genes, *cryaa*, *cyp3a65*, *gnat2*, *otpa*, *rp1l1b*, *tyrp1a* and *vsx1* were down-regulated in selenium-stressed embryos at 24 hpf, as determined by qPCR (Appendix A). In addition, down-regulated expression of *otpa* at 24 hpf was also observed in selenium-stressed embryos by WISH (Appendix A). The expression of *lama1*, *opn1sw1*, *pde6b*, *pde6ha*, *rbp3*, *rx1* and *rx2* increased in selenium-stressed embryos at 24 hpf (Appendix A). Furthermore, the expression of nearly all the tested genes above decreased except *lama1*, *otpa*, *otx2* and *tyrp1a* in selenium-stressed embryos at 96 hpf (Figure 1C). The expression of *opn1sw1*, *cryaa* and *pde6a* was also verified by WISH, which was consistent with the qPCR results (Figure 1D). We also tested the expression of other genes marked ganglion, muller glia, horizontal, bipolar and amacrine cells in retina at 96 hpf. Among these tested genes, the expression of *alcama*, *cahz2*, *gfap*, *mdka*, *isl1a*, *neurod1* and *pax6a* were up-regulated, while the expression of *atoh7*, *gja9a*, *lin7a* and *th* were down-regulated in selenium-stressed embryos at 96 hpf (Appendix A).

### 3.2. Excessive Selenium Caused ROS Accumulation and Oxidative Stress in Embryonic Cells

To dissect the underlying mechanism for selenium-induced eye malformation, the intracellular ROS levels in the selenium-stressed embryos were measured at 96 hpf. The results revealed that the treated embryos had massive accumulation of ROS (Figure 2A). Then, the expression of oxidative stress marker genes (*cox4i2*, *hmox2*, *lox* and *nrf2*) was determined by qPCR in both groups at 24 hpf and 96 hpf. The oxidative stress marker genes *cox4i2*, *hmox2* and *lox* were up-regulated in the selenium-stressed group at 24 hpf (Figure 2B1), while down-regulated in selenium-treated embryos at 96 hpf (Figure 2B2). There was no change in the expression of *nrf2* at 24 and 96 hpf. WISH data revealed the expression of *cox4i2* and *nrf2* was consistent with the result of qPCR, and the supplementation of GSH could not recover the expression of *cox4i2* (Figure 2C). The slightly down-regulated expression of *cox4i2*, *lox*, *nrf2* and significantly up-regulated expression of *hmox2* was also detected by RNA-seq (Appendix A). Furthermore, the activity of antioxidant and redox enzymes was also tested in selenium-stressed embryos. The selenium-stressed group showed significant increases in MDA content (Figure 2D1) and GR activity (Figure 2D2) compared with the control group, but a decrease in GSH content (Figure 2D3) at 96 hpf. GSH supplementation recovered the MDA (Figure 2D1) and GSH content to normal level, but did not change GR activity (Figure 2D2) in selenium-stressed embryos. The GR activity (Figure 2D2) and GSH content (Figure 2D3) were reduced by NAC supplementation, but MDA content had no change in selenium-stressed embryos supplemented with NAC (Figure 2D1).

### 3.3. Excessive Selenium Triggered Cell Apoptosis and Proliferation in Zebrafish Eyes

To reveal the types of cell death, we tested the apoptosis in zebrafish embryonic eyes at 48, 72 and 96 hpf. TUNEL assay demonstrated the activation of apoptosis in selenium-stressed embryonic eyes (Figure 3A,D1 and Appendix A), and the number of apoptosis was gradually increased from 48 to 96 hpf in selenium-stressed embryos. The apoptosis marker protein Caspase3 exhibited an increased expression in selenium-stressed embryonic retina at 96 hpf (Figure 3B,D2). The qPCR results revealed that the expression of *apaf1*, *caspase3a*, *caspese7*, *caspase9* and *tp53* was up-regulated in the treatment groups at 24 hpf (Figure 3E1). However, the expression of *caspase7* and *tp53* was normal at 96 hpf (Figure 3E2). The slightly up-regulated expression of *apaf1*, *caspase3a*, *caspase9* and *tp53* was also detected by RNA-seq (Appendix A). To further analyze the induction of eye defects, cell proliferation was also examined. The results demonstrated a significant increase in the number of phospho-H3-positive nuclei in selenium-stressed zebrafish embryonic eyes at 96 hpf (Figure 3C,D3).

### 3.4. Excessive Selenium-Induced Ferroptosis in Eye Cells

Transmission electron microscopy (TEM) revealed that the structure of organelle mitochondria in selenium-stressed embryonic eye cells exhibited the typical characteristics of ferroptosis [36,37], including an increase in mitochondrial membrane density, a decrease in transmittance, disappearance of mitochondria crista and rupture of outer membrane (Figure 4A). Since iron overload is a major characteristic of ferroptosis, we also measured the level of cytosolic iron, and representative images revealed that the selenium-stressed embryonic eyes had obviously higher iron levels in cytoplasm at 96 hpf, as identified with the FerroOrange probes. GSH supplementation could reduce the iron level, but NAC induced more serious iron overload in selenium-stressed embryos at 96 hpf (Figure 4B).

Next, to investigate the mechanism for ferroptosis, the key genes in ferroptosis, including *acsl4a*, *acsl4b*, *gpx4a*, *gpx4b* and *slc7a11*, were examined by qPCR in the control and selenium-stressed embryos at 24 hpf and 96 hpf. Among the tested genes, *gpx4a* and *gpx4b* were significantly up-regulated, while *slc7a11* was significantly down-regulated in selenium-stressed embryos at 24 hpf (Figure 4C1). The expression of *acsl4a*, *acsl4b*, *gpx4a* and *gpx4b* was significantly decreased, and that of *slc7a11* showed no significant change in selenium-stressed embryos at 96 hpf relative to the control embryos (Figure 4C2). In order to further explore the role of ferroptosis-related genes in the early eye developmental defects, ten DEGs (*acsl1b*, *acsl5*, *cp*, *fthl27*, *hmox1a*, *hpx*, *map1lc3cl*, *meltf*, *tfa* and *tfr1b*) related to ferroptosis screened with the RNA-seq data were also tested by qPCR in selenium-stressed embryos (Appendix A). The expression of most genes including *acsl1b*, *cp*, *fthl27*, *hmox1a*, *hpx*, *map1lc3cl*, *meltf*, *tfa* and *tfr1b* was up-regulated, but *acsl5* showed no significant change in selenium-treated embryos relative to those in the control embryos at 24 hpf (Appendix A). Furthermore, the expression of *acsl1b*, *acsl5*, *hpx*, *meltf* and *tfr1b* was down-regulated, while *cp*, *fthl27* and *map1lc3cl* were up-regulated in selenium-stressed embryos at 96 hpf (Appendix A).

### 3.5. Scavenging of ROS Failed to Rescue Selenium-Induced Eye Defects

Two ROS scavengers (NAC and GSH) were chosen to rescue the selenium-induced microphthalmia defects. The measurement of eye radius showed that neither GSH nor NAC restored the eye deformity, and NAC even aggravated the deficient phenotype (Figure 5A,B). qPCR and WISH were performed to examine the expression levels of genes related to eye development in embryos treated by selenium + NAC and selenium + GSH at 96 hpf. As a result, GSH supplementation up-regulated the expression of *gnat2* and *opn1sw2*, but down-regulated that of *rbp3*, while led to no significant change in *cyp3a65*, *pde6a* and *rp1l1b* expression in the selenium-stressed embryos at 96 hpf (Figure 5C). However, nearly all the tested genes were down-regulated in embryos treated by selenium + NAC at 96 hpf (Figure 5C). The WISH data of *pde6a* were consistent with the results of qPCR (Figure 5D). GSH supplementation could not adequately rescue the expression of *opn1sw1* according to WISH (Figure 5D).

### 3.6. Promotion of Ferroptosis and Apoptosis Rescued the Eye Defects in Selenium-Stressed Embryos

A ferroptosis inhibitor ferrostatin-1 (Fer-1) and a ferroptosis and apoptosis activator cisplatin (CDDP) were used to rescue the selenium-induced eye defects. As a result, supplementation of Fer-1 did not rescue the selenium-induced microphthalmia defects (Figure 6A,B). qPCR results revealed that the supplementation of Fer-1 up-regulated *gnat2*, *opn1sw1* and *pde6a* but down-regulated *cyp3a65*, while showing no obvious effect on the expression of *rbp3* and *rp1l1b* in selenium-stressed embryos at 96 hpf (Figure 6C). However, CDDP could rescue the selenium-caused microphthalmia defects to almost normal levels (Figure 6A,B), and its supplementation increased the expression of *cyp3a65, gnat2*, *opn1sw1*, *pde6a*, *rbp3* and *rp1l1b* in selenium-stressed embryos at 96 hpf (Figure 6D).

### 3.7. NAC, GSH, Fer-1 and CDDP Altered the Expression of Genes Related to Apoptosis and Ferroptosis

The expression of apoptosis-related genes (*apaf1* and *caspase3a*) and ferroptosis-related genes (*acsl4a* and *gpx4a*) were also detected by qPCR to investigate the effect of NAC, GSH, Fer-1 or CDDP supplementation on the expression of these key genes in selenium-stressed zebrafish embryos at 96 hpf. As illustrated in Figure 7, GSH supplementation could recover the expression of *apaf1*, *caspase3a* and *acsl4a* but decreased *gxp4a* in selenium-stressed embryos at 96 hpf. NAC supplementation increased the expression of *caspase3a* and reduced the expression of *gpx4a* in selenium-stressed embryos at 96 hpf. Fer-1 was found to up-regulate ferroptosis-related *acsl4a* and apoptosis-related *apaf1* in selenium-stressed embryos at 96 hpf. CDDP supplementation recovered the expression of *caspase3a*, *acsl4a* and *gpx4a* to nearly the normal levels in selenium-stressed embryos. *apaf1* was also recovered by CDDP supplementation.

## 4. Discussion

### 4.1. Toxic Effect of Excessive Selenium on Zebrafish Eyes

Industrial and agricultural activities are dominant anthropogenic sources of selenium pollution, which poses serious threats to water systems [38]. In recent years, the toxicity of selenium has been widely reported. Chronic exposure to high concentrations of selenomethionine can impair the social and antipredator behaviors of adult zebrafish [39]. Parental life cycle exposure to waterborne selenite can retard follicle maturity and increase progeny deformity in zebrafish [40]. In addition, excessive selenite causes cardiac and neural defects in zebrafish embryos, and this embryotoxicity can be alleviated by folic acid [41]. Although the effects of selenium toxicity have been well documented, the phenotypes and mechanisms specific to an organ are not fully understood. The present study investigated the toxic effects of selenium on the eyes of zebrafish embryos by identifying the phenotypes and determining the expression of genes involved in eye development.

Selenium-stressed zebrafish embryos exhibited smaller eye radius at 48, 72 and 96 hpf, suggesting impairment of embryonic eye development (Figure 1A). H&E staining analysis of the eyes at 96 hpf demonstrated obvious tissue damage (Figure 1B). Moreover, the retinal cells of selenium-stressed embryos showed condensed mitochondria at 96 hpf (Figure 4A), suggesting that selenium may induce ocular defects by affecting a mitochondrial pathway. The expression of nearly all genes related to eye development was down-regulated in selenium-stressed embryos at 96 hpf (Figure 1C,D). Among the tested genes, *cryaa* (encoding αA-crystallin) plays an important role in the development of the lens. Mutation in the *cryaa* gene can cause congenital cataracts, either alone or in association with other pathological conditions such as microcornea and microphthalmia [42,43]. The decrease in the expression of *cryaa* indicated that selenium affects zebrafish lens development. Down-regulated expression of *cyp3a65* indicated that selenium might affect eye development of zebrafish embryos by disturbing retinol metabolism. Selenium-stressed embryos showed a decrease in the mRNA level of the rod-related gene *rho* at 96 hpf. The *opn1sw1* and *gnat2* genes, which are essential for the development of cone cells, were both down-regulated at 96 hpf, while *opn1sw1* was up-regulated at 24 hpf. The photoreceptor cell-specific phosphodiesterase 6 (PDE6) subfamily consists of three catalytic subunits encoded by the *pde6a*, *pde6b* and *pde6c* genes. The catalytic activity of PDE6 is regulated by the inhibitory subunits encoded by *pde6g* and *pde6h*. The *pde6a* and *pde6b* genes, which give rise to a catalytic heterodimer, and the inhibitory subunit gene *pde6g* were expressed in rod cells, whereas *pde6c*, which results in a catalytic homodimer, and the inhibitory subunit gene *pde6h* were expressed in cone cells [44,45]. The expression of all the tested PDE6 genes decreased in selenium-stressed embryos at 96 hpf. The decrease in the expression of *rho*, *opn1sw1, gnat2, pde6a*, *pde6b*, *pde6c* and *pde6ha* genes in selenium-stressed embryos suggested that photoreceptors (including cone cells and rod cells) are sensitive to selenium exposure. Retinol binding protein 3 (RBP3), a retinol transport protein mainly secreted by the photoreceptors, was up-regulated at 24 hpf, while down-regulated at 96 hpf in selenium-stressed embryos. Partial retinal photoreceptor loss was reported to be associated with a decrease in the transcription of interphotoreceptor RBP3 in a transgenic mouse model [46]. Retinitis pigmentosa-1-like-1 (RP1L1) is a component of the photoreceptor axoneme, the backbone structure of the light-sensing outer segment of photoreceptors. Mutation of *rp1l1* was found to result in progressive photoreceptor dysfunction and age-related macular degeneration-like pathology in zebrafish [47]. The expression of rod and cone genes is usually down-regulated during retinal degeneration, including age-related macular degeneration (AMD), cone dystrophy and retinitis pigmentosa [48,49]. Moreover, the progenitor genes *rx1* and *rx2* of eyes showed decreases in expression in selenium-stressed embryos at 96 hpf. We also tested the expression of other genes marked ganglion (*alcama* and *atoh7*), muller glia (*cahz2* and *gfap*), horizontal (*gja9a* and *mdka*), bipolar (*vsx1*, *isl1a* and *lin7a*) and amacrine (*neurod1*, *pax6a* and *th*) cells in retina at 96 hpf. The abnormal expression of all the tested genes indicated that excessive selenium poses a great threat to the eye development of zebrafish, particularly to photoreceptors.

### 4.2. Effects of Excessive Selenium on Apoptosis, Proliferation and Ferroptosis of Embryonic Eye Cells

To identify the mechanism of the selenium-induced eye defects, we investigated the changes in intracellular ROS level and redox status in selenium-stressed embryos at 96 hpf. An increase in ROS was observed in the eyes of selenium-stressed embryos (Figure 2A). ROS accumulation can cause serious damage to cell structures and ROS-induced lipid peroxidation produces MDA [50]. In this study, the MDA production was significantly increased in selenium-stressed zebrafish embryos (Figure 2D1), indicating that selenium induces oxidative stress in zebrafish embryos. GSH provides the reducing equivalent and is a critical component in cellular antioxidant system. Here, a decrease in GSH level was detected in selenium-stressed embryos, which might be a stress response to the toxicity of excessive selenium (Figure 2D3). Oxidative stress is believed to occur due to an imbalance in the biological oxidant-to-antioxidant ratio, which can be ascribed to the increase in free radical generation and decrease in the activity of antioxidants [51]. In order to maintain a redox homeostasis, antioxidants such as GR and some detoxification responses may be initially activated in eye cells in response to selenium toxicity. GR serves mainly to maintain the cellular homeostasis of GSH/GSSG ratio or the redox balance of the cell. An increase in GR activity (Figure 2D2) was detected in selenium-stressed embryos at 96 hpf, which may be related to the reduction in GSH (Figure 2D3). Cytochrome *c* oxidase (COX) is the terminal oxidase in the mitochondrial electron transport chain, which catalyzes the reduction in oxygen molecules by reducing cytochrome *c*. *cox4i2* is responsible for encoding one subunit of COX [51]. A decrease in COX activity can promote ROS production. Heme oxygenase-2 (*hmox2*) can convert heme to biliverdin, which is then transformed into a strong antioxidant bilirubin by biliverdin reductase [52]. Lysyl oxidase (*lox*) is an enzyme in the cell redox system; LOX up-regulation was reported to enhance oxidative stress [53,54]. Nuclear factor (erythroid-derived 2)-like 2 (*nrf2*) is an important nuclear transcription factor for the whole antioxidant defense system. Under stress, it is involved in initiating the transcription of many antioxidative enzymes, such as superoxide dismutase (SOD), catalase and glutathione S-transferase (GSTP) [55]. The expression of these four genes was up-regulated in selenium-stressed embryos at 24 hpf, indicating that the antioxidant defense system is responsive to selenium stress. However, the down-regulation of *cox4i2*, *hmox2* and *lox* at 96 hpf revealed that excessive selenium inhibited the expression of these genes and led to decreases in related functional proteins and an increase in ROS, suggesting the exacerbation of oxidative stress.

Apoptosis is a kind of programmed cell death which can effectively remove excess and injured cells in the body and maintain the stability of tissues and organs. The inhibition of the apoptotic pathway has been reported to be closely associated with the occurrence of many cancers [56,57]. The interaction of multiple apoptosis-related factors generates apoptosis-inducing signals, which are transmitted to the initiator caspases (cysteinyl aspartate specific proteases); the caspases are then activated and begin to cleave cellular proteins to result in apoptosis [58]. Therefore, caspases play a core role in cell apoptosis [59,60]. The apoptotic peptidase activating factor 1 (*apaf1*) is required for mitochondrial pathways of apoptosis [61]. *tp53* is an important and widespread tumor suppressor gene, which is closely related to tumorigenesis and cell apoptosis [62]. Under stress, *tp53* will be significantly activated, inducing the cells to enter the apoptosis process and eliminating the cells with cancerous tendency. The expression of the tested genes *apaf1*, *caspase3a* and *caspase9* was up-regulated in selenium-stressed embryos at 24 hpf and 96 hpf. *caspase7* and *tp53* were also up-regulated in selenium-stressed embryos at 24 hpf. These results indicate that selenium began to induce apoptosis at an early stage in the embryos (Figure 3E). However, cell apoptosis was increased from 48 to 96 hpf by TUNEL assay (Figure 3A,D1 and Appendix A) in selenium-stressed embryonic eyes, which might be due to the exacerbation of oxidative stress in the embryos at 96 hpf. Our results demonstrated a significant change in the number of phospho-H3-positive nuclei in selenium-stressed embryonic eye cells at 96 hpf (Figure 3C,D3). The two antagonistic reactions (cell apoptosis and proliferation) seem to be induced by the same extracellular stimulus. Therefore, further research is needed to fully explore whether proliferation is induced directly by selenium or as a compensation response for selenium-induced cell death.

Ferroptosis is also a type of programmed cell death dependent on iron and characterized by the accumulation of lipid peroxides. It is currently believed that ferroptosis is involved in the occurrence and development of various diseases, such as tumorigenesis, ischemia-reperfusion injury (IRI), renal failure, nervous system diseases and blood system diseases [63]. Here, typical ferroptotic characteristics including condensed mitochondria and iron overload were also detected in selenium-stressed embryonic eye cells at 96 hpf (Figure 4A,B). Moreover, the increase in MDA content and decrease in GSH in selenium-stressed embryos at 96 hpf also indicated the occurrence of ferroptosis (Figure 2D1,D3). *acsl4* and *gpx4* are well-known genes that positively and negatively regulate ferroptosis, respectively [64]. The expression of the *gpx4a* and *gpx4b* genes was significantly up-regulated in selenium-stressed embryos at 24 hpf, while down-regulated at 96 hpf (Figure 4C). The expression pattern of the *gpx4a* and *gpx4b* genes is similar to that of oxidative stress markers. The *acsl4* gene showed no obvious change in selenium-treated embryos at 24 hpf, but was down-regulated at 96 hpf (Figure 4C), indicating the inhibition of ferroptosis [65]. System XC- is a heterodimer composed of SLC7A11 and SLC3A2. The down-regulated expression of *slc7a11* inhibits the cystine uptake through system XC-, resulting in a decrease in cystine-dependent GPXs activity and thereby leading to ferroptosis. *p53* is a transcription inhibitor of SLC7A11 and participates in the process of ferroptosis [66]. *slc7a11* was down-regulated in selenium-stressed embryos at 24 hpf (Figure 4C1), which might be inhibited by the elevated activity of Gpx4. Cell apoptosis seems to be closely associated with ferroptosis under selenium stress.

### 4.3. Effects of NAC, GSH, Fer-1 and CDDP on Selenium-Induced Zebrafish Embryonic Microphthalmia Defects

In order to identify the potential therapeutic targets of eye diseases related to selenium for future study, two ROS scavengers (GSH and NAC), a ferroptosis inhibitor (Fer-1) and a ferroptosis and apoptosis activator (CDDP) were selected to test their effects on selenium-induced zebrafish embryonic eye defects. The results show that NAC supplementation aggravated eye defects in selenium-stressed embryos at 96 hpf. In addition, it down-regulated the expression of ferroptosis-related gene *gpx4a* and eye developmental genes *cyp3a65*, *opn1sw2*, *pde6a*, *rbp3* and *rp1l1b*, and up-regulated the expression of apoptosis-related gene *caspase3a* in selenium-stressed embryos at 96 hpf (Figure 5 and Figure 7). The results of MDA content, GSH content, GR activity and FerroOrange staining in the embryos treated by selenium + NAC also suggested that NAC might trigger more serious ferroptosis (Figure 2D,4B), which might be due to the synergistic toxic effects of NAC and excessive selenium on zebrafish embryonic eyes. The supplementation of GSH and Fer-1 showed similar rescuing effects on the selenium-induced defects, which is consistent with the findings of previous studies [67,68]. Both GSH and Fer-1 could increase the expression of eye developmental genes *gnat2* and *opn1sw2* (or *opn1sw1*) and decrease that of some genes such as *rbp3* or *cyp3a65*. However, the phenotype of microphthalmia was still not fully rescued under GSH and Fer-1 supplementation in selenium-stressed embryos at 96 hpf (Figure 5 and Figure 6A–C). The genes associated with ferroptosis and apoptosis were also tested to explore how GSH and Fer-1 regulate the two pathways. The results showed that GSH and Fer-1 could up-regulate the *apaf1* and *acsl4a* genes (Figure 7) in the selenium-stressed embryos at 96 hpf, suggesting that the regulation of apoptosis and ferroptosis might be an effective way to rescue the eye defects induced by excessive selenium.

Compared with non-malignant cells, cancer cells (especially tumor stem cells) are strongly dependent on the micronutrient iron for growth. This iron dependency can make cancer cells more vulnerable to iron-catalyzed cell death, which is referred to as ferroptosis. Ferroptosis has already been reported to be a potentially promising way to kill therapy-resistant cancer cells [69]. CDDP, an alkylating chemotherapeutic agent, is widely used for many cancer therapies such as sarcoma, gonad, breast, lung, bladder and lymphoma1 cancers [70,71]. It can result in the formation of DNA adducts, the production of ROS and increases in lipid peroxidation and mitochondrial stress. However, it can also cause many adverse effects including ototoxicity, gonadotoxicity, nephrotoxicity, neurotoxicity and marrow suppression [72,73]. Selenium has been reported to have a protective effect against cisplatin-related retinotoxicity [74]. In this study, CDDP supplementation restored both the phenotype and the expression of the tested eye marker genes to nearly normal levels in selenium-stressed embryos at 96 hpf (Figure 6A,B,D). Moreover, CDPP supplementation recovered the expression of apoptosis- and ferroptosis-related genes in selenium-stressed embryos, suggesting that apoptosis and ferroptosis might be a way for animals to deal with excessive selenium. However, more research is needed to fully explore how CDPP recovered the selenium-induced eye defects. One possibility is that apoptosis and ferroptosis are molecular mechanisms by which embryos respond to excessive selenium; the earlier activation of ferroptosis and apoptosis by trace CDDP may be helpful to maintain the normal development of embryos under selenium stress.

## 5. Conclusions

This study made the first attempt to illustrate the potential mechanism for the effect of excessive selenium on eye development during zebrafish embryogenesis. Our results revealed that excessive selenium induces microphthalmia defects and impairs the structure of zebrafish retinal cells. Specifically, excessive selenium induces cell apoptosis and ferroptosis through oxidative stress, leading to zebrafish eye defects. Moreover, we found that the CDDP could rescue the selenium-induced eye defects. This study may facilitate a better understanding of the cellular toxicity of excessive selenium and its role in eye diseases.

## Figures and Tables

**Figure 1 ijms-23-04783-f001:**
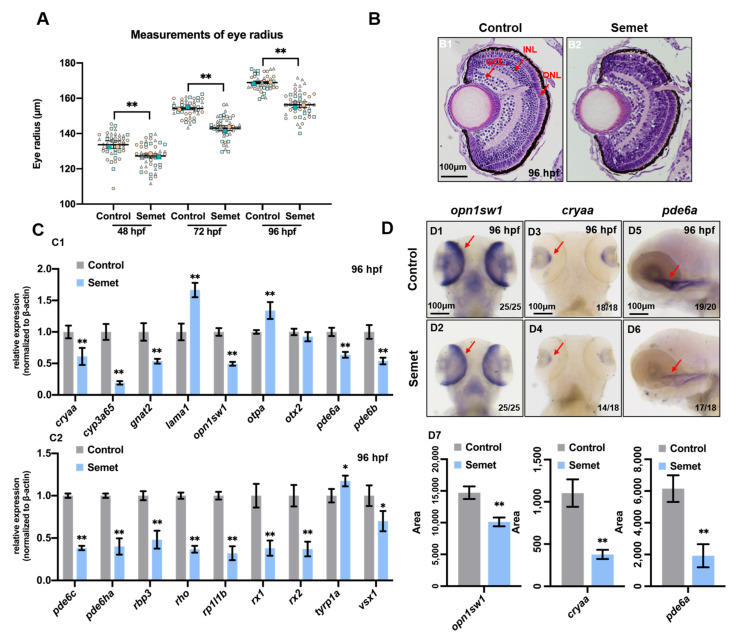
Eye developmental defects in selenium-treated embryos. (**A**) Measurement of eye radius of embryos from control and selenium-treated groups at 48 hpf, 72 hpf and 96 hpf. (**B**) H&E staining analysis of embryonic eyes from control and selenium-treated groups at 96 hpf (red arrows indicate GCL, INL and ONL layer of retina). (**C**) Expressions of eye marker genes in embryos from control and selenium-treated groups at 96 hpf (**C1**,**C2**). (**D**) WISH data of *opn1sw1*, *cryaa* and *pde6a* in embryos from control and selenium-treated groups at 96 hpf (**D1**–**D6**). Quantification analysis of the WISH data in different samples (**D7**). **, *P* < 0.01; *, *P* < 0.05.

**Figure 2 ijms-23-04783-f002:**
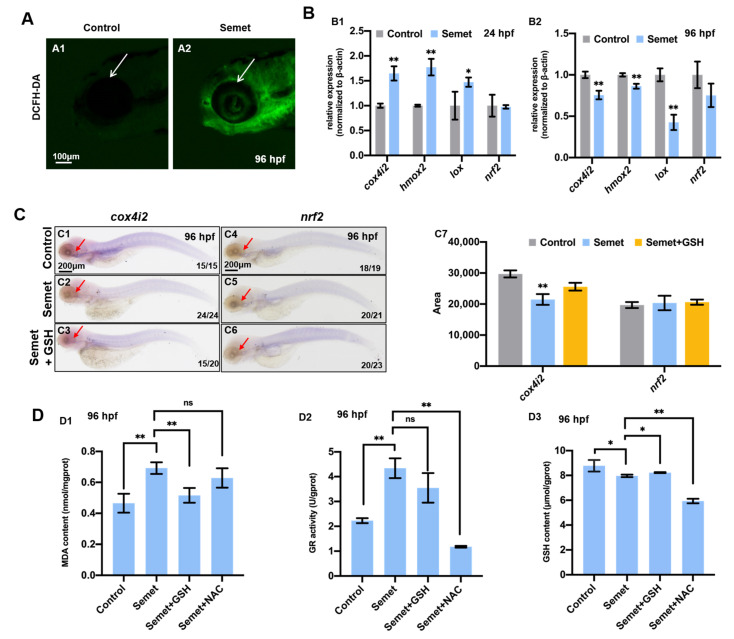
Oxidative stress in selenium-treated embryos. (**A**) ROS levels in control and selenium-treated embryos at 96 hpf were indicated by DCFH-DA staining. (**B**) Expression of oxidative stress marker genes in control and selenium-treated embryos at 24 hpf (**B1**) and 96 hpf (**B2**). (**C**) WISH data of *cox4i2* and *nrf2* in embryos from control, semet and semet plus GSH groups at 96 hpf (**C1**–**C6**). Quantification analysis of the WISH data in different samples (**C7**). (**D**) MDA content (**D1**), GR activity (**D2**) and GSH content (**D3**) in control, semet, semet plus GSH and semet plus NAC embryos at 96 hpf. **, *P* < 0.01; *, *P* < 0.05.

**Figure 3 ijms-23-04783-f003:**
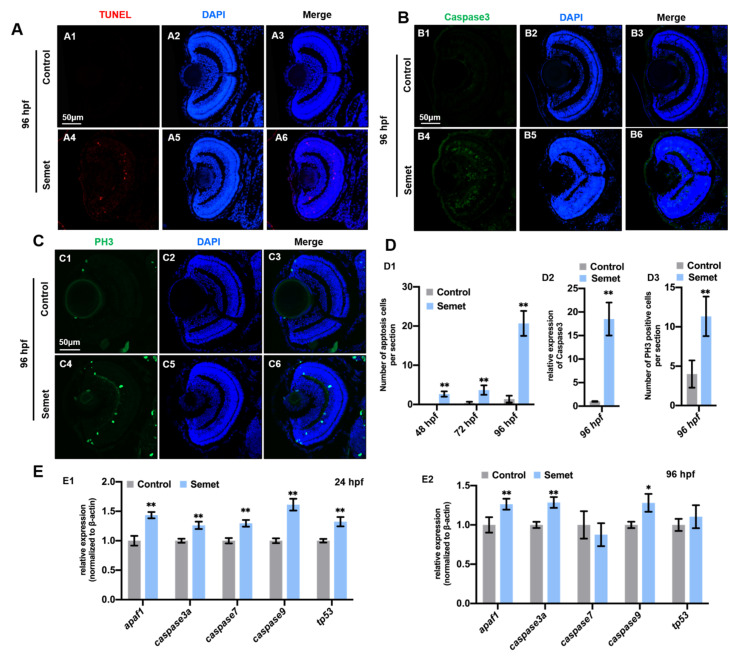
Cell apoptosis and proliferation in selenium-treated embryos. (**A**) Cell apoptosis assay by TUNEL (red dots) detection in section of control (**A1**–**A3**) and selenium-treated (**A4**–**A6**) embryos at 96 hpf. (**B**) Immunostaining of Caspase3 in section of control (**B1**–**B3**) and selenium-treated (**B4**–**B6**) embryos at 96 hpf. (**C**) Cell proliferation assay by PH3 staining (green dots) in section of control (**C1**–**C3**) and selenium-treated (**C4**–**C6**) embryos at 96 hpf. (**D**) Number of apoptosis cells (**D1**), relative expression of caspase3 (**D2**) and number of PH3 positive cells (**D3**). (**E**) Expression of apoptosis marker genes in control and selenium-treated embryos at 24 hpf (**E1**) and 96 hpf (**E2**). **, *P* < 0.01; *, *P* < 0.05.

**Figure 4 ijms-23-04783-f004:**
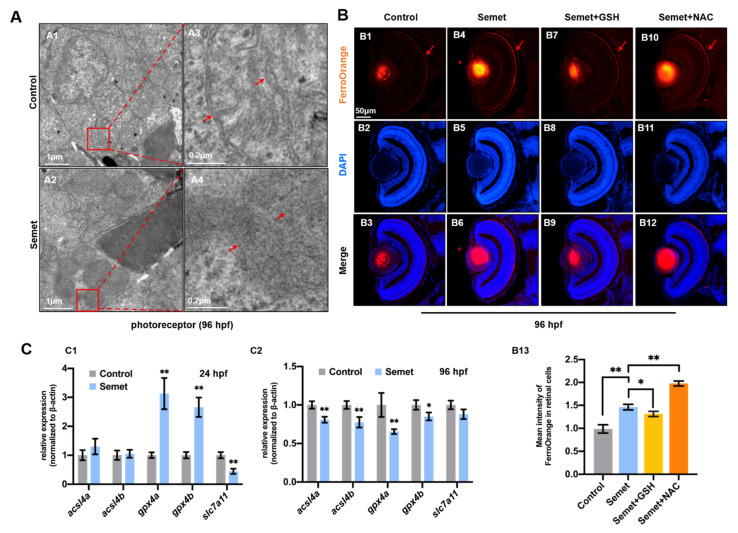
Ferroptosis in selenium-treated embryos. (**A**) TEM analysis of retinal cells in control and selenium-stressed embryos at 96 hpf (red squares and arrows indicate mitochondria). (**A3**,**A4**) are magnified domains of red boxes marked in (**A1**,**A2**), respectively. (**B**) Iron distribution assay by FerroOrange probe in control, semet, semet plus GSH and semet plus NAC groups at 96 hpf (**B1**–**B12**). Measurement of FerroOrange staining in retinal cells (**B13**). (**C**) Expressions of ferroptosis marker genes in control and selenium-treated embryos at 24 hpf (**C1**) and 96 hpf (**C2**). **, *P* < 0.01; *, *P* < 0.05.

**Figure 5 ijms-23-04783-f005:**
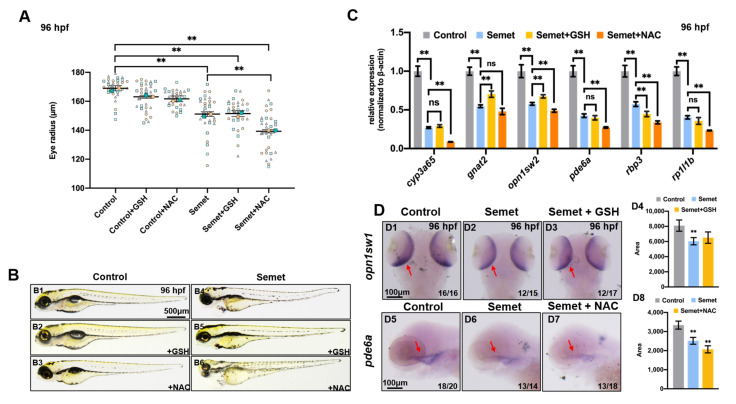
Effect of ocular defects with addition of ROS scavenger GSH and NAC. (**A**) Measurement of eye radius of embryos from control, semet, semet plus GSH and semet plus NAC groups at 96 hpf. (**B**) Phenotypes of representative control, semet, semet plus GSH and semet plus NAC groups at 96 hpf. (**C**) The expression of eye marker genes in control, semet, semet plus GSH and semet plus NAC groups at 96 hpf. (**D**) WISH data of *opn1sw1* (**D1**–**D3**) and *pde6a* (**D5**–**D7**) in control, semet, semet plus GSH and semet plus NAC groups at 96 hpf. Quantification analysis of the WISH data of *opn1sw1* (**D4**) and *pde6a* (**D8**) in different samples. **, *P* < 0.01; *, *P* < 0.05.

**Figure 6 ijms-23-04783-f006:**
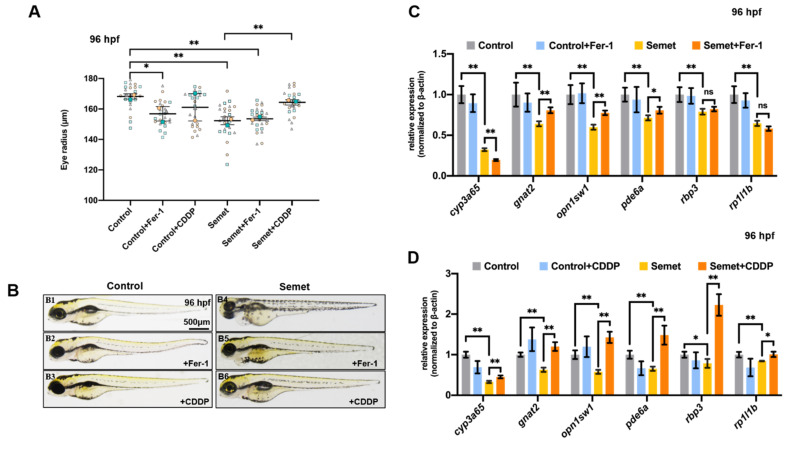
Effect of ocular defects with addition of ferroptosis inhibitor Fer-1 and activator CDDP. (**A**) Measurement of eye radius of embryos from control, semet, semet plus Fer-1 and semet plus CDDP groups at 96 hpf. (**B**) Phenotypes of representative control, semet, semet plus Fer-1 and semet plus CDDP groups at 96 hpf. (**C**) The expression of eye marker genes in control, semet and semet plus Fer-1 groups at 96 hpf. (**D**) The expression of eye marker genes in control, semet and semet plus CDDP groups at 96 hpf. **, *P* < 0.01; *, *P* < 0.05.

**Figure 7 ijms-23-04783-f007:**
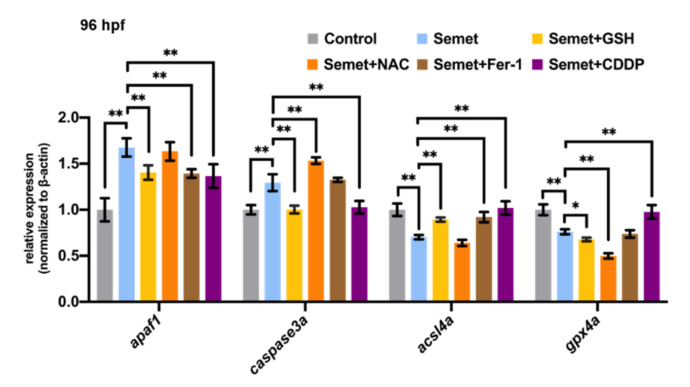
Effects of GSH, Fer-1, NAC and CDDP on ferroptosis and apoptosis in selenium-treated embryos at 96 hpf. **, *P* < 0.01; *, *P* < 0.05.

## Data Availability

Non applicable.

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
