# Peer review of "Ferroptosis and Apoptosis Are Involved in the Formation of L-Selenomethionine-Induced Ocular Defects in Zebrafish Embryos"

_ijms, 2022, doi:10.3390/ijms23094783_

Round 1

Reviewer 1 Report

Rereview of manuscript from Gao et al.

Thank you for making so many revisions despite all of the pandemic related difficulties you are experiencing. Overall the figures are much clearer as is the text.

There are just a couple of things to still clarify.

  1. Methods are greatly improved but still do not contain the stage at when the embryos are added to the 0.5µM selenomethionine. Please add how old the embryos were when they were treated.
  2. Finally, below I suggest slight rephrasing of the abstract for clarity

Selenium is an essential trace element for humans and other vertebrates, playing important roles in antioxidant defense, neurobiology and reproduction. However, the toxicity of excessive selenium has not been thoroughly evaluated, especially for the visual system of vertebrates.

In this study, fertilized zebrafish embryos were treated with 0.5 μM L-selenomethionine to investigate how excessive selenium alters zebrafish eye development. Selenium-stressed zebrafish embryos showed microphthalmia altered expression of genes required for retinal neurogensis. Moreover, ectopic proliferation, disrupted mitochondrial morphology, elevated ROS-induced oxidative stress, apoptosis, and ferroptosis were observed in selenium-stressed embryos. Two antioxidants -- reduced glutathione (GSH) and N-acetylcysteine (NAC) – and the ferroptosis inhibitor ferrostatin (Fer-1) were unable to rescue selenium-induced eye defectsbut the ferroptosis and apoptosis activator cisplatin (CDDP) was able to improve microphthalmia.and the expression of retina-specific genes in selenium-stressed embryos.

In summary, our results reveal that ferroptosis and apoptosis might play a key role in selenium

induced defects of embryonic eye development. The findings will not only provide new insights

into the selenium-induced cellular damage and death, but also important implications for studying

the association between excessive selenium and ocular diseases in the future.

Author Response

1. Methods are greatly improved but still do not contain the stage at when the embryos are added to the 0.5µM selenomethionine. Please add how old the embryos were when they were treated.

Response: Sorry for the mistake, we have added this content in the revised manuscript (line106 - line107).

2.Finally, below I suggest slight rephrasing of the abstract for clarity

Response: Thanks for your suggestion, we have revised the abstract.

Reviewer 2 Report

All the issues previously indicated have been properly corrected and all the suggested experiments have been performed. In this context, because this article addresses a relevant public health issue and the results are new, I recommend its publication in IJMS.

Author Response

Thanks for your review again.